# CB_1_ Ligand AM251 Induces Weight Loss and Fat Reduction in Addition to Increased Systemic Inflammation in Diet-Induced Obesity

**DOI:** 10.3390/ijms231911447

**Published:** 2022-09-28

**Authors:** Lannie O’Keefe, Teresa Vu, Anna C. Simcocks, Kayte A. Jenkin, Michael L. Mathai, Deanne H. Hryciw, Dana S. Hutchinson, Andrew J. McAinch

**Affiliations:** 1Institute for Health and Sport, Victoria University, P.O. Box 14428, Melbourne, VIC 8001, Australia; 2Drug Discovery Biology, Monash Institute of Pharmaceutical Sciences, Monash University, Parkville, VIC 3052, Australia; 3School of Science, Western Sydney University, Campbelltown, NSW 2560, Australia; 4The Florey Institute of Neuroscience and Mental Health, Parkville, Melbourne, VIC 3052, Australia; 5School of Environment and Sciences, Griffith University, Nathan, QLD 4111, Australia; 6Griffith Institute for Drug Discovery, Griffith University, Nathan, QLD 4111, Australia; 7Australian Institute for Musculoskeletal Science (AIMSS), Victoria University, Melbourne, VIC 8001, Australia

**Keywords:** diet-induced obesity (DIO), skeletal muscle (SM), endocannabinoid system (ECS), AM251, adiponectin, adipose tissue, cannabinoid receptor 1 (CB_1_)

## Abstract

Diet-induced obesity (DIO) reduces fatty acid oxidation in skeletal muscle and decreases circulating levels of adiponectin. Endocannabinoid signaling is overactive in obesity, with some effects abated by antagonism of cannabinoid receptor 1 (CB_1_). This research aimed to determine if treatment with the global CB_1_ antagonist/inverse agonist, AM251, in high-fat diet (HFD) fed rats influenced adiponectin signaling in skeletal muscle and a “browning” of white adipose tissue (WAT) defined by UCP1 expression levels. Male Sprague Dawley rats consumed an HFD (21% fat) for 9 weeks before receiving daily intraperitoneal injections with vehicle or AM251 (3 mg/kg) for 6 weeks. mRNA expression of genes involved in metabolic functions were measured in skeletal muscle and adipose tissue, and blood was harvested for the measurement of hormones and cytokines. Muscle citrate synthase activity was also measured. AM251 treatment decreased fat pad weight (epididymal, peri-renal, brown), and plasma levels of leptin, glucagon, ghrelin, and GLP-1, and increased PAI-1 along with a range of pro-inflammatory and anti-inflammatory cytokines; however, AM251 did not alter plasma adiponectin levels, skeletal muscle citrate synthase activity or mRNA expression of the genes measured in muscle. AM251 treatment had no effect on white fat UCP1 expression levels. AM251 decreased fat pad mass, altered plasma hormone levels, but did not induce browning of WAT defined by UCP1 mRNA levels or alter gene expression in muscle treated acutely with adiponectin, demonstrating the complexity of the endocannabinoid system and metabolism. The CB_1_ ligand AM251 increased systemic inflammation suggesting limitations on its use in metabolic disorders.

## 1. Introduction

The endocannabinoid system (ECS) is a lipid signaling system involved in the regulation of appetite, fatty acid oxidation, glucose metabolism, and inflammation [1,2,3]. Energy homeostasis regulation via the ECS works through both central and peripheral mechanisms [4]. There is evidence from both in vitro and in vivo studies implicating the ECS in a series of pathological conditions [5]. Antagonism of cannabinoid receptor 1 (CB_1_), which is upregulated in obesity, promotes weight loss and decreases obesity-associated co-morbidities [6]. The use of the global CB_1_ antagonist Rimonabant in the treatment of obesity has been hampered by its adverse effects on the central nervous system (CNS) resulting in adverse psychological side effects [7]. 

Skeletal muscle plays an important role in whole-body energy metabolism including fatty acid oxidation and glucose homeostasis [5]. An increase in intramyocellular fatty acids due to decreased oxidative capacity in skeletal muscle is thought to result in cellular stress and contribute to the low-grade chronic inflammation observed in obesity [6]. Antagonism of CB_1_ in skeletal muscle upregulates the expression of genes involved in fatty acid oxidation and glucose control [6], thereby having a beneficial effect on skeletal muscle function in obesity. Additionally, the adipokine adiponectin can act on skeletal muscle adiponectin receptors (AdipoR1, and AdipoR2) [8] to positively affect insulin-stimulated glucose transporter expression and glucose uptake [8]. CB_1_ antagonism increases plasma adiponectin levels independent of weight loss [9], whether this increase in adiponectin correlates with improved adiponectin signaling in skeletal muscle is unknown. 

In addition to the role of the ECS in skeletal muscle, elucidating the mechanisms of adipocyte dysfunction in obesity and its links to the ECS may be important in identifying additional novel anti-obesity treatment targets. The global CB_1_ inverse agonist/antagonist AM251 has been shown to reduce weight gain in animal models of obesity by contributing to a reduction in food intake [10] and an increase in energy expenditure [11]. 

The effect of AM251 to reduce white adipose tissue (WAT) weight in high fat diet (HFD) fed models is thought to be mainly a result of increased sympathetic tone to WAT depots, while AM251 can increase uncoupling protein 1 (UCP1) mRNA and protein levels in brown adipose tissue (BAT) [12]. It is unclear whether there is any “browning” of WAT depots with AM251; white adipocytes from CB1 knockout mice have increased UCP1 mRNA levels associated with increased cellular respiration [12], which could indicate that CB1 antagonism could directly result in WAT browning. 

Therefore, we investigated in a diet induced obesity (DIO) rat model (1) the physiological effects of the CB_1_ ligand, AM251, (2) the inflammatory profile of AM251 treatment (3) the effect of AM251 treatment on specific adipocyte genes in both WAT and BAT depots and (4) the effects of AM251 treatment and acute adiponectin exposure on fatty acid oxidative pathways in skeletal muscle. The hypothesis of this study is that AM251 will influence adiponectin signaling pathways in skeletal muscle and contribute to “browning” of WAT.

## 2. Results

### 2.1. AM251 Decreases Fat Pad Weight and Alters Plasma Hormone and Cytokine Levels in an HFD Model

Male rats were placed on an HFD for 9 weeks and then allocated to vehicle or AM251 treatment for a further 6 weeks. No differences in weight (Vehicle 582.9 ± 14.35 g, AM251 576.0 ± 11.51 g), food intake and body composition (data not shown) were detected between groups at 9 weeks prior to the treatment period. As we have previously published in this cohort, AM251 caused less weight gain (Vehicle 655.9 ± 19.23 g compared to AM251 607.2 ± 12.43 g; *p* ≤ 0.05), and a transient reduction in food consumption (data not shown) [13]. In agreeance with reductions in weight gain, there was a significant decrease in epididymal, peri-renal and brown fat pads (Table 1; *p* ≤ 0.05), with no changes in the heart and liver weights (Table 1).

While we have previously demonstrated no change following AM251 treatment in glucose homeostasis as measured by Glucose Tolerance Tests (GTTs) and Insulin Tolerance Tests (ITTs) [13] we were interested in whether AM251 altered plasma levels of adiponectin, leptin, glucagon, PAI-1, GLP-1, and ghrelin. While AM251 did not alter plasma adiponectin levels (Figure 1a), ghrelin, leptin, glucagon, and GLP-1 were significantly decreased, and PAI-1 levels were significantly increased with AM251 treatment (Figure 1b–e; *p* ≤ 0.05). 

We measured circulating inflammatory markers to assess whether AM251 treatment could alter these following an HFD. In HFD rats, compared to the vehicle treated rats, AM251 treatment significantly increased plasma levels of EPO, IFN-γ, IL-1α, IL-1β, IL-2, IL-4, IL-6, IL-12p70, IL-17α, IL-18, and RANTES (Table 2; *p* < 0.05). In HFD rats, AM251 treatment did not alter plasma G-CSF, GM-CSF, GRO/KC, IL-5, IL-10, IL-13, MCSF, MCP-1, MIP-1α, MIP-3α, TNF-α or VEGF levels (Table 2).

### 2.2. The Effect of AM251 on Skeletal Muscle in an HFD Rat Model

There was no change in skeletal muscle citrate synthase (CS) activity in either red (Figure 2a) or white gastrocnemius (Figure 2b) muscle of animals treated for six weeks with AM251, compared to vehicle control. 

The treatment with AM251 for six weeks in HFD rats resulted in a non-significant trend to increase PGC-1α mRNA expression in the EDL (Table 3; *p* = 0.066). No significant changes in mRNA were observed in either the EDL or soleus muscle of HFD animals following six weeks of treatment with AM251 (Table 3). 

To assess whether acute adiponectin exposure can alter the expression of several genes involved in skeletal muscle function in rats on an HFD with or without treatment for six weeks with AM251, isolated soleus and EDL muscle was incubated in the presence or absence of adiponectin (0.1 µg/mL for 30 min). In soleus muscle, despite non-significant trend for adiponectin exposure in AM251 treated animals to alter AdipoR1 (*p* = 0.097) no changes in mRNA were observed in either of the HFD groups (Table 3). In EDL muscle, despite non-significant trends for adiponectin exposure in AM251 treated animals to alter FAT/CD36 (*p* = 0.067) and PGC-1α (*p* = 0.094) no changes in mRNA were observed in either of the HFD groups (Table 3).

### 2.3. Effect of AM251 on Adipose Tissue Gene Expression

AM251 reduced weight gain in rats placed on an HFD [13], which was associated with changes in fat depot weights (Table 1). Therefore, we investigated whether AM251 affected the expression of genes involved in fat function/identification including the brown fat markers UCP1, PRDM16 and CPT1b, and the white fat markers HOXC9 and TCF21. Adrb3 mRNA was measured since β_3_-ARs are important in adipose tissue function in response to sympathetic stimuli, and CB_1_/CB_2_ (Cnr1 and Cnr2) mRNA was also measured. Glucose transporter 1 and 4 (Slc2a1 and Slc2a4) expression was also measured in the different adipose tissue depots.

In epididymal WAT (eWAT), AM251 in HFD rats caused a significant decrease in Adrb3, Cnr1, Slc2a4, HOXC9 and TCF21 (Figure 3; *p* ≤ 0.05) mRNA expression, with no changes in Cnr2, IL-1β, IL-6 or TNF-α mRNA levels. There was no evidence for browning of eWAT depots in HFD rats since UCP1 mRNA levels did not alter, and other markers for BAT function (CPT1b, PRDM16) also decreased (Figure 3; *p* ≤ 0.05).

In peri-renal WAT (pWAT), AM251 in HFD rats also caused a significant decrease in CPT1b and PRDM16 (Figure 4; *p* ≤ 0.05) while no changes were detected in UCP1, IL-1β or IL-6 mRNA expression. Slc2a1, and TCF21 (Figure 4; *p* ≤ 0.05) mRNA expression were also decreased following AM251 treatment in HFD rats. 

In BAT, AM251 treatment for six weeks in HFD rats caused a significant decrease in the expression of Adrb3, CPT1b, Slc2a4, PRDM16, UCP1, IL-1β and leptin (Figure 5; *p* ≤ 0.05). Cnr1 and Cnr2 was undetectable in BAT (data not shown).

## 3. Discussion

Despite advances in research, limited pharmacological treatments are available for obesity and its co-morbidity. Antagonism of CB_1_ induces weight loss leading to a reduction in various co-morbidities [1]. The current study demonstrated that 6 weeks of treatment with the global CB_1_ inverse agonist/antagonist, AM251, in DIO rats clearly influences whole body physiological parameters including decreased fat pad weight, and, as we have previously demonstrated decreased food intake and weight gain [13]. At the tissue level, specific changes were observed in the expression of several genes in adipose tissue depots, but no changes were observed in skeletal muscle. These results demonstrate the complex interplay between the endocannabinoid system and metabolic tissues in obesity and weight loss. However, the administration of AM251 increased systemic inflammation with increased levels of a wide range of cytokines. 

The browning of WAT depots is thought to be an avenue for therapeutic approaches to obesity [14]. Our hypothesis was that AM251 may increase browning of WAT that contributes to its actions on lowering body weight and reduced fat mass. However, AM251 treatment in HFD rats did not support this hypothesis in eWAT and pWAT, as there was a decrease in PRDM16 and CPT1b mRNA expression and no alteration in UCP1 mRNA expression. This is despite evidence that CB_1_ knockout mice have WAT displaying increased mitochondrial function and elevated UCP1 levels [12], and chronic treatment of dogs with rimonabant results in the increased expression of brown fat markers (including UCP1) in WAT [12]. An upregulation of β_3_-ARs in WAT is thought to be partly responsible for the browning of WAT by rimonabant [15]. Our results show that AM251 treatment was accompanied by a downregulation in β_3_-AR mRNA expression. Since β_3_-ARs are important in lipolysis and lowering body fat levels [16], this may suggest that the lowering of body fat by AM251 may be independent of β_3_-ARs in WAT.

BAT is an important regulator of whole-body energy homoeostasis and involved in thermoregulation of rodents [16], and its presence/function in humans has led BAT to be a new target for human obesity treatments [16]. In DIO mice treated with rimonabant, improvements in glucose utilisation, increased energy expenditure and activation of BAT independent on weight loss were reported [17]. In our DIO rat model, AM251 treatment caused a decrease in Adrb3, UCP1, CPT1b, GLUT4, PRDM16, and leptin mRNA levels in BAT. It’s important to note that changes in UCP1 mRNA levels do not necessarily reflect changes in UCP1 protein levels [16] and that UCP1 still needs to be activated by a stimuli such as sympathetic stimulation of β3-ARs. In our AM251 treated DIO rats, there was however a decrease in Adrb3 which was coupled with a decrease in UCP1 as well as CPT1b and PRDM16 mRNA levels, which may suggest a blunting of thermogenic responses in these animals. However, functional readouts in the future would enable a better conclusion to be made. 

The decrease in plasma leptin in our DIO model following AM251 treatment agrees with previous research in mice following 15-day treatment with rimonabant [18]. Ghrelin which is implicated to stimulate appetite [13] was decreased following six weeks of treatment with AM251, and supports our previous observations in this cohort of a decrease in food intake, albeit only transiently [13]. In further agreement, glucagon which has been demonstrated to be reduced following weight loss [13] also decreased following the treatment period. In contrast to these observed changes in leptin, ghrelin, glucagon, and PAI-1 we did not observe any change in plasma adiponectin levels. This disagrees with previous research in animals and humans that have demonstrated that antagonists/inverse agonists of CB_1_ increase plasma adiponectin levels [1]. 

While no changes in plasma adiponectin were observed in the current study, we were interested in whether there was any alteration in some key genes involved in adiponectin signaling and muscle oxidative capacity, following treatment with AM251 in HFD fed rats and whether these genes were altered following acute exposure to adiponectin. It has previously been demonstrated in human primary skeletal muscle myotubes derived from both lean and obese individuals that AM251 treatment increases AMPKα1 and decreases PDK4 mRNA expression, while no changes were observed in the expression of AMPKα2 [6]. In the current study, despite the reduction in weight gain and alteration in adipose tissue mass, we did not observe any changes in mRNA expression following AM251 treatment in key genes involved in adiponectin signaling or alteration in PDK4. However, in agreement with the human primary skeletal muscle culture work [6,19] we did not observe any changes in a key gene involved in mitochondrial biogenesis, PGC-1α, which is supported by the lack of change in citrate synthase activity in the current study.

We have previously demonstrated in human primary skeletal muscle myotubes that 6 h of treatment with 0.1 µg/mL of globular adiponectin (gAd) increased AMPK activity in myotubes derived from lean individuals but only the higher dose of 0.5 µg/mL of gAd increased AMPK activity derived from individuals with obesity and obesity and diabetes [20]. The current study demonstrates in rodents that acute, 30-min exposure to adiponectin following six weeks of an HFD did not alter AMPKα2 mRNA in the soleus or EDL. We have also previously demonstrated that 6 h of adiponectin treatment in human primary myotubes derived from lean individuals increases AdipoR1 but not AdipoR2 mRNA expression, whereas myotubes derived from individuals who had obesity or obesity and diabetes did not alter AdipoR1 or AdipoR2 [21]. In agreement with this work, whether the HFD animals were treated with AM251 or not, acute exposure to adiponectin did not alter AdipoR1 or AdipoR2 in the soleus or EDL. While overexpression of Appl1 has been demonstrated to increase adiponectin signaling [22] we also did not see any compensatory increase in the mRNA expression of Appl1 in the animals following adiponectin treatment. While the lack of change in PGC-1α supports our previous work in human primary myotubes following gAd treatment [21,23], it is possible that the acute ex vivo exposure to adiponectin was not long enough to enable the detection of any subsequent mRNA changes and future work should increase the ex vivo exposure time. 

AM251 treatment of HFD fed rats caused an increase in several plasma pro-inflammatory cytokines: namely IFN-γ, IL-1α, IL-1β, IL-2, IL-4, IL-6, IL-12p7, IL-17α, IL-18, and RANTES. Adipose tissue remodelling during weight loss can initiate an inflammatory response [24], and may at least partially explain the increased levels of circulating cytokines observed in the current study. IL-6 has also been observed to increase following weight reduction [25]. However, in contrast to these studies, other circulating cytokines have been shown to decrease when reducing adiposity [26] and a higher dose of AM251 (10 mg/kg) for 4 weeks reduced pro-inflammatory M1 adipose tissue macrophages in the presence of a reduction in fat pad mass [27]. It is known that some cytokines such as IL-18 may not be affected by weight loss but with alterations in insulin resistance [28]. IL-18 however, increased in the current study after AM251 treatment. Interestingly plasma adiponectin levels have been demonstrated to be negatively correlated with IL-6 and IL-18 [29]. Thus, it is possible that the increases in these cytokines because of AM251 treatment negated the expected increase in plasma adiponectin as a result of decreased adipose tissue mass. Rimonabant treatment for 12 weeks in obese women with polycystic ovarian syndrome resulted in increased circulating levels of VEGF however, the treatment did not alter TNF-α, IL-1β, IL-2, and IL-6 [30], which is in contrast to our results following AM251 treatment. IL-10, which has been previously demonstrated to correlate with circulating adiponectin levels [31], did not change in the current study. In view of these collective results, we measured the mRNA expression of three of these cytokines (IL-1β, IL-6 and TNF-α) in the different fat depots to determine if the source of the inflammation was adipose tissue (we were not able to detect sufficient amounts of TNF-α in pWAT and BAT for quantification). Taken together our observed decrease in IL-1β in BAT and lack of alterations of this and IL-6 and TNF-α in other adipose tissue depots and previously observed reductions in IL-6 and TNF-α in adipose tissue macrophages following treatment with AM251 [27] suggest that the source of increased circulating cytokines in this study does not appear to be derived from adipose tissue. Indeed a recent paper [32] has indicated that CB1 deficiency in non-immune cells promotes DIO resistance, while CB1 deficiency in immune cells further worsens DIO induced inflammation. This may indicate that blockade of the CB1 receptor pharmacologically has both beneficial and detrimental effects on DIO phenotypes. Both rimonabant and AM251 are effective in reducing/preventing obesity in a number of rat nd mouse models as reviewed previously [33]. Structurally, they are similar, differing only in a single halogen substituent. Both are potent and selective CB1 receptor antagonists, and can both act as inverse agonists at higher concentrations. However, there are differences in their selectivity profiles, with AM251 being ~1000 fold more selective for the mouse CB1 vs. the mouse CB2, as opposed to rimonabant being only 25 fold more selective [34]. There are also reported differences in their off-target activities, with Soethoudt et al. [34] showing that AM251 had activity at 16 off-target receptors including GPR55, whereas rimonabant only had 8 off-target activities. An additional level of complexity may arise from ligand biased signaling, which is the ability of different ligands to differentially activate different receptor signaling effectors/responses. Ligand biased signaling at the CB1 receptor has been reported [35] although to our knowledge there are no studies comparing bias of AM251 vs. rimonabant. Given that research into AM251 and other CB1 antagonists/inverse agonists is still continuing, it will be important for future studies to determine if there are critical differences between these ligands and rimonabant, especially its effect on adverse psychiatric events.

## 4. Materials and Methods

### 4.1. Animals and Experimental Protocol 

All animal experimental procedures were approved by the Howard Florey Animal Ethics Committee (AEC 11-036), which operates under the guidelines of the National Health and Medical Research Council of Australia. Seven-week-old male Sprague Dawley rats were used (The Animal Resource Centre, Canning Vale, WA, Australia). Following a 1–2 week acclimatization period, the rats were individually housed in a plastic tub with stainless steel lid (cage dimensions; width 27.5 × length 41.0 × height 25.5 cm) (R.E. Walters, Sunshine, Victoria, Australia) in an environmentally controlled laboratory (ambient temperature 22–24 °C) with a 12 h light/dark cycle (07:00–19:00).

### 4.2. Rodent Model of Diet-Induced Obesity and AM251 Pharmacological Treatment

Following the acclimatization period rats receive a high-fat diet (HFD) containing 40% digestible energy from lipids (Specialty Feeds SF00-219, Glen Forrest, WA, Australia) for 9 weeks, as described in our previously published study [13]. Throughout the study, animals could access food and water ad libitum. Animals were then maintained on the HFD and treated for a further six weeks with a daily i.p. injection, with either vehicle (0.9% isotonic saline solution containing 0.75% Tween 80: n = 9–10), or 3 mg/kg body weight of AM251 (Cayman Chemicals, Ann Arbour, MI, USA) dissolved in the vehicle solution (n = 9–10). Following treatment, rats were anesthetized with 3% isoflurane inhalation (Abbott Laboratories, Illinois, United States) with each animal undergoing surgical removal of skeletal muscle, cardiac blood was then collected confirming death, with all other major organs including fat pads were removed post-mortem, weighed, and stored at −80 °C for further analyses.

### 4.3. Physiological Measurements

Total body weight (g) and food consumption (g/day) were recorded daily throughout the entire treatment period. Food pellets at the top of the cage and any remaining visible spillage within the cage were collected and weighed to determine total daily food consumption. 

### 4.4. Muscle Sample Preparations 

Each muscle was carefully dissected into longitudinal strips from tendon to tendon using a 27-gauge needle. Following dissection, the gastrocnemius muscle was separated into red and white muscle and snap-frozen in liquid nitrogen for subsequent measurement of citrate synthase activity. Immediately after their removal the longitudinal strips of the soleus and EDL were incubated in an oxygenated organ bath (37 °C pre-gassed at 95% O_2_ −5% CO_2_). Each longitudinal strip was placed in an individual chamber of Krebs-Henseleit buffer (Sigma-Aldrich, St. Louis, MO, USA) with 0.1 µg/mL of adiponectin (Sigma-Aldrich, St. Louis, MO, USA) or the contralateral hind limb muscle exposed to Vehicle for 30 min. Muscle strips were taken from opposing limbs from each animal to ensure that alternate muscles were exposed to the treatment. Following incubation, the muscle samples were snap frozen in liquid nitrogen and stored at −80 °C for further analyses. 

### 4.5. RNA Extraction

Skeletal muscle total RNA was isolated from the soleus and EDL muscle by using TRIzol Reagent (Invitrogen, Carlsbad, CA, USA) as previously described [21]. In short, rodent muscle tissue extracts (approximately 15 mg) were dissociated using 1000 mg of ceramic/silica beads and RNA was extracted in TRIzol and treated with the RQ1 RNase–free DNase kit (Promega Corporations, Madison, WI, USA) following instructions of the manufacturer. First-strand cDNA was then generated from 0.3 µg of template RNA using the iScript™ cDNA synthesis kit (Bio-Rad Laboratories, Hercules, CA, USA) using random hexamers and oligo dTs. cDNA was stored at −20 °C for subsequent analysis. qPCR was conducted using iQ™ SYBR Green Supermix (Bio-Rad Laboratories, Hercules, CA, USA) and the MyiQ™ single colour qPCR detection system (Bio-Rad Laboratories, Hercules, CA, USA). qPCR reactions were run for 50 cycles of 95 °C for 15 s and 60 °C for 60 s. Samples of mRNA were amplified to test skeletal muscle for expression of genes: AdipoR1, AdipoR2, APPL1, APPL2, FAT/CD36, AMPK, PDK4 and PGC-1α (Geneworks, Thebarton, SA, USA). The data was normalized to housekeeping genes, GAPDH, cyclophilin and β-actin (Table 4).

Adipose RNA was isolated using according to the manufacturer’s protocol with some modifications to remove the fat, according to previously published protocols [36]. Approximately 100 mg of frozen adipose tissue was placed into 1 mL of Tri reagent (Invitrogen, Melbourne, Australia), containing 1 g of sterile ceramic beads, and then homogenized using the Fast prep FP 120 (Thermo Electron Corporation) for 20 s at 5.5 Hz. Samples were centrifuged at 13,000× *g* at 4 °C for 15 min the supernatant was recovered leaving the top fat layer. The supernatant was centrifuged again at 12,000× *g* at 4 °C for 10 min. The lysate was recovered from under the fat layer into a new sterile tube. This was repeated a total of 3 times to ensure all the fat was removed from the lysate. 200 µL chloroform was added to the tubes, quickly vortexed, and left on ice for 5 min and centrifuged at 13,000× *g* for 15 min at 4 °C. The upper clear layer was transferred to 500 µL isopropanol (Sigma-Aldrich, MO, USA) and 10 µL 5M NaCl. The samples were stored at −20 °C for a minimum of 2 h or overnight to precipitate the RNA. Samples were then centrifuged (13,000× *g*, 4 °C, 20 min), the supernatant removed leaving a white pellet (RNA), washed in 400 µL 75% ethanol and centrifuged (9000× *g*, 8 min 4 °C). The supernatant was removed, and the RNA pellet air-dried in a fume hood for 10 min to evaporate remaining ethanol. Heated 60 °C diethylpyrocarbonate (DEPC) water (Thermo-Fisher, Waltham, MA, USA) was added (10 µL), samples were vortexed and placed on ice. 1 µL of the 10 µL was removed and diluted in 19 µL of DEPC treated water for the determination of RNA concentration. First-strand cDNA was then generated from 0.3 µg of template RNA using iScript™ cDNA synthesis kit (Bio-Rad Laboratories, Hercules, CA, USA). qPCR was performing on a LightCycler 480 (Roche, NSW, Australia), as follows: initial heating to 50 °C for 2 min, then 95 °C for 10 min, before each cycle consisted of 95 °C for 15 s and 60 °C for 2 min for 40 cycles, then all samples were cooled to 25 °C. In adipose tissue, samples of mRNA were amplified to test the expression of Brown adipocyte genes: UCP1, PRDM16, CPT1B; WAT specific genes: HOXC9, TCF21; Receptors; CB_1_, CB_2_, β_3_ adrenoceptor; Transporters: GLUT1, GLUT4 (ThermoFisher Scientific, Waltham, MA, USA; Table 5). The data was normalized to hypoxanthine phosphoribosyltransferase 1 (HPRT1).

### 4.6. Citrate Synthase Analysis

Citrate Synthase (CS) activity in red and white gastrocnemius muscle was assayed as previously described [37]. Briefly, after the addition of 5 µL of muscle homogenate (0.175 mM KCl & 2 mM EDTA, pH 7.4), 230 µL of reagent cocktail ((3 mM Acetyl CoA (Sigma A-2056), 100 mM Tris buffer (BioRad, Hercules, CA, USA); pH 8.3), 1 mM DTNB (Sigma D-8130) and 15 µL of 10 mM Oxalacetate (Sigma 0-4126). CS activity was measured at 412 nm (xMark™ Microplate Spectrophotometer, BioRad). Enzyme activity was expressed relative to wet weight, change in absorbance/min was calculated to measure CS activity [38].

### 4.7. Plasma Hormone and Cytokine Analysis

Cardiac blood was extracted at the time of death and transferred to EDTA tubes (McFarlane Medical, Victoria, Australia), and processed as previously described [39]. Plasma samples were prepared following the manufacturer’s instructions for analysis of diabetes 5-plex panel and the rat cytokine 23-Plex panel multiplex protein arrays (BioRad, BioRad Laboratories, Munich, Germany). The diabetes 5-plex panel consisted of the following measurements: 1. Ghrelin; 2. Leptin; 3. Glucagon; 4. PAI-1 (*Plasminogen activator inhibitor-1*); 5. GLP-1 (*Glucagon-like peptide-1*). The rat cytokine 23-Plex panel kit consisted of the following measurements: 1. EPO (*Erythropoietin*), 2. G-CSF (*Granulocyte Colony Stimulating Factor*), 3. GM-CSF (*Granulocyte-Macrophage Colony Stimulating Factor*), 4. GRO/KC (*Growth-related oncogene*), 5. IFN-γ (*Interferon Gamma*), 6. IL-α (*Interleukin 1-alpha*), 7. IL-β (*Interleukin 1- beta*), 8. IL-2 (*Interleukin 2*), 9. IL-4 (*Interleukin 4*), 10. IL-5 (*Interleukin 5*), 11. IL-6 (*Interleukin 6*), 12. IL-10 (*Interleukin 10*), 13. IL-12p70 (*Interleukin 12*), 14. IL-13. (Interleukin 13), 15. IL-17α (*Interleukin 17*), 16. IL-18 (*Interleukin 18*), 17. M-CSF (*Macrophage colony-stimulating factor*), 18. MCP-1 (*Monocyte Chemotactic Protein 1*), 19. MIP-3α (*Macrophage Inflammatory Protein 3α*), 20. RANTES, 21. TNF-α (*Tumour Necrosis Factor Alpha*), 22. VEGF (*Vascular Endothelial Growth Factor*). MIP-1α (*Macrophage Inflammatory Protein 1α)* did not work for any sample and thus was excluded from the analysis. Plasma adiponectin was analyzed according to manufactures instructions (AdipoGen, Liestal, Switzerland).

### 4.8. Statistical Analysis

Real-time semi-quantitative PCR (qPCR) values are presented as arbitrary units mean ± SEM, normalized to housekeeping genes and expressed as 2^−^^Δ^^ΔCT^ in arbitrary units. GraphPad 8.0 Prism software, all data are presented as mean ± SEM. Analysis of the groups was determined using an unpaired or paired t-test as appropriate, where data was not normally distributed a non-parametric Mann–Whitney or Wilcoxon matched-pairs test was completed. Significance was accepted when *p* ≤ 0.05.

## 5. Conclusions

The data from the current study demonstrate the changes in vivo and ex vivo following treatment with the global CB_1_ inverse agonist/antagonist, AM251. This study has shown reduced epididymal, peri-renal and brown fat pad mass, and circulating levels of leptin, glucagon, ghrelin, and GLP-1. In skeletal muscle, AM251 treatment for six weeks, or when the muscle from these animals were exposed to ex vivo acute treatment with adiponectin, did not alter mRNA expression of genes associated with adiponectin signaling or oxidative capacity. We also saw no evidence for the browning of WAT in chronically treated AM251 rats on an HFD, which suggests that browning of WAT is not a mechanism contributing to the anti-obesity effects of AM251. We also observed that AM251 treatment resulted in an increase in pro-inflammatory cytokines, representing whole body systemic inflammation. Thus, despite AM251 having a positive impact on obesity with respect to food intake, weight loss and fat pad weight, it results in systemic inflammation bringing into question its continued use in research. 

## Figures and Tables

**Figure 1 ijms-23-11447-f001:**
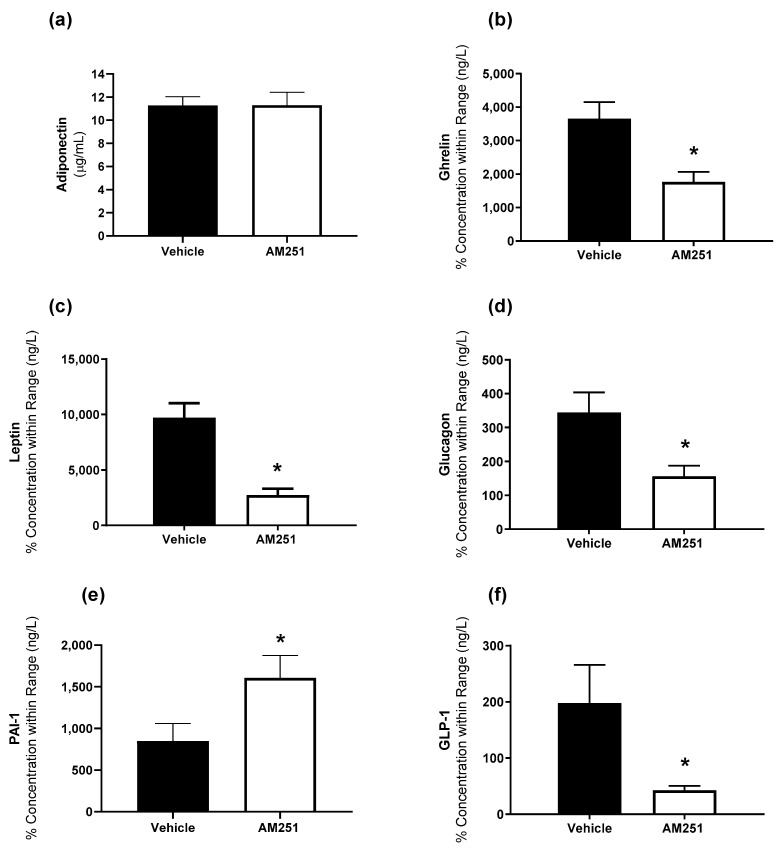
The effect of AM251 treatment on hormones involved in energy homeostasis. HFD rats were injected daily with either AM251 (3 mg/kg of body weight, ip) or vehicle for six weeks. Plasma levels of (**a**) adiponectin, (**b**) ghrelin, (**c**) leptin, (**d**) glucagon, (**e**) PAI- 1 and (**f**) GLP-1. All results are presented as a mean ± SEM from n = 9 (AM251) and n = 5–9 (vehicle) treated rats on an HFD. Significance * *p* ≤ 0.05 compared to vehicle.

**Figure 2 ijms-23-11447-f002:**
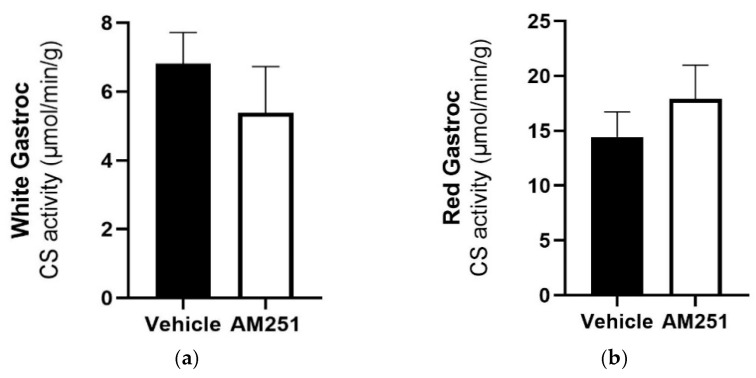
The effect of chronic AM251 treatment on citrate synthase activity in red and white gastrocnemius skeletal muscle in HFD rats. HFD rats were injected daily with either AM251 (3 mg/kg of body weight, ip) or vehicle for six weeks. (**a**) Red gastrocnemius (n = 9) and (**b**) White Gastrocnemius (n = 9) citrate synthase activity following treatment with AM251 compared to vehicle treated rats.

**Figure 3 ijms-23-11447-f003:**
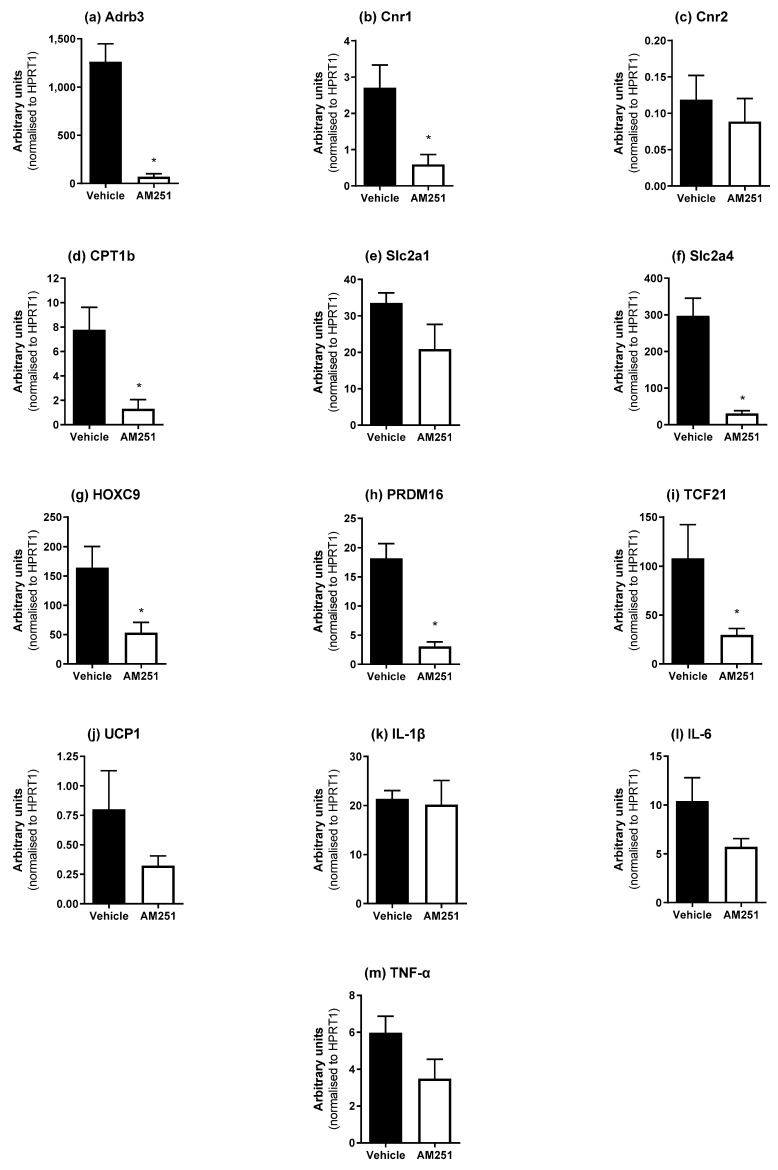
Effect of AM251 treatment in HFD rats on epididymal WAT mRNA expression. HFD rats were injected daily with either AM251 (3 mg/kg of body weight, ip) or vehicle for six weeks. All results are presented as a mean ± SEM in arbitrary units (normalized to housekeeping gene HPRT1) from n = 6–9 (AM251) and n = 7–9 (vehicle) treated rats on an HFD. Significance * *p* ≤ 0.05 compared to vehicle.

**Figure 4 ijms-23-11447-f004:**
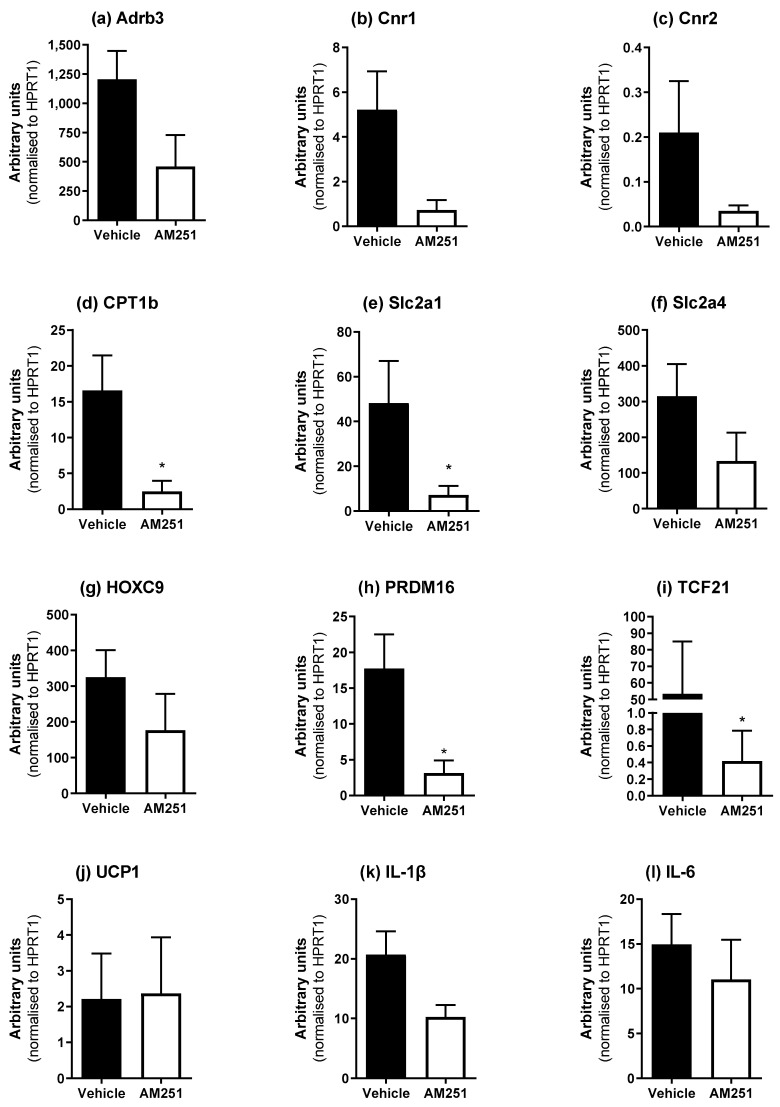
Effect of AM251 treatment in HFD rats on peri-renal WAT mRNA expression. HFD rats were injected daily with either AM251 (3 mg/kg of body weight, ip) or vehicle for six weeks. All results are presented as a mean ± SEM in arbitrary units (normalized to housekeeping gene HPRT1) from n = 4 (AM251) and n = 8–10 (vehicle) treated rats on an HFD. Significance * *p* ≤ 0.05 compared to vehicle.

**Figure 5 ijms-23-11447-f005:**
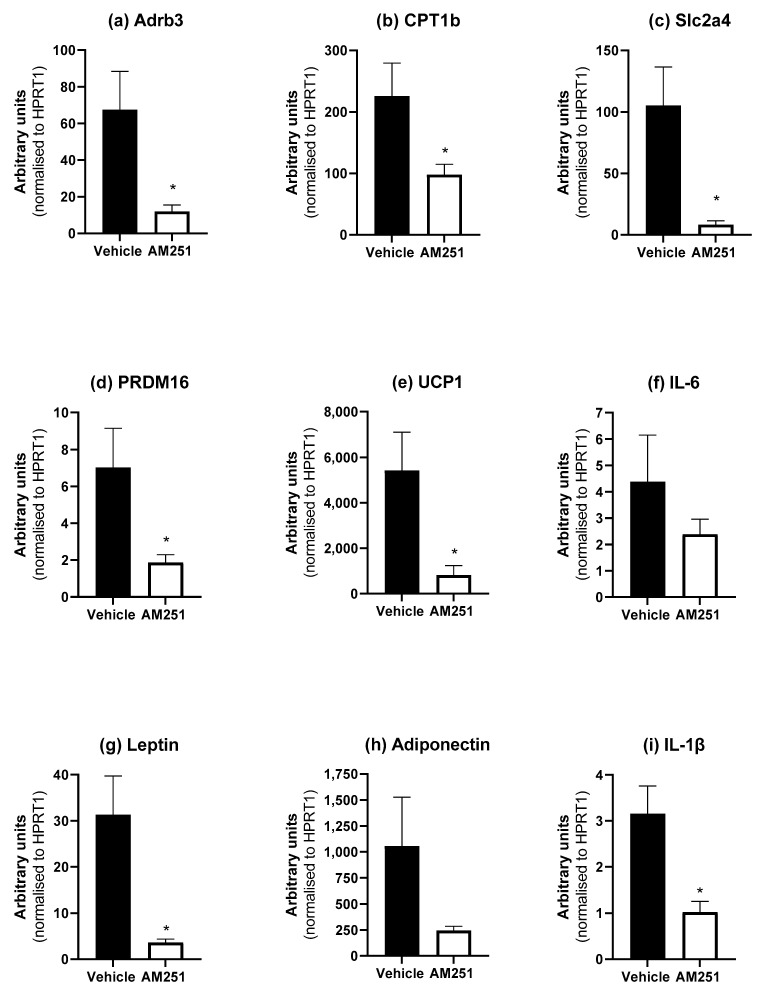
Effect of AM251 treatment in HFD rats on BAT mRNA expression. HFD rats were injected daily with either AM251 (3 mg/kg of body weight, ip) or vehicle for six weeks. All results are presented as a mean ± SEM in arbitrary units (normalised to housekeeping gene HPRT1) from n = 6–10 (AM251) and n = 9 (vehicle) treated rats on an HFD. Significance * *p* ≤ 0.05 compared to vehicle.

**Table 1 ijms-23-11447-t001:** The effect of AM251 treatment on organ weights (g). HFD fed rats were injected daily with either AM251 (3 mg/kg of body weight, ip) or vehicle for six weeks. All results are presented as a mean ± SEM from n = 10 (AM251) and n = 6–10 (vehicle) treated rats on an HFD. Significance * *p* ≤ 0.05 compared to vehicle.

Organ Weight (g)	Vehicle	AM251
Heart	1.60 ± 0.08	1.61 ± 0.04
Liver	22.27 ± 0.84	20.79 ± 0.76
Epididymal Fat Pad	10.32 ± 0.77	6.73 ± 0.82 *
Peri-renal Fat Pad	11.90 ± 1.15	8.50 ± 1.13 *
Brown Fat Pad	1.08 ± 0.10	0.66 ± 0.07 *

**Table 2 ijms-23-11447-t002:** Plasma cytokine levels following AM251 treatment in HFD rats. HFD rats were injected daily with either AM251 (3 mg/kg of body weight, ip) or vehicle for six weeks. All results are presented as a mean ± SEM from n = 6–9 (AM251) or n = 5–10 (vehicle) treated rats on an HFD in % concentration within range (ng/L). Significance * *p* ≤ 0.05 compared to vehicle.

Cytokine	Vehicle	AM251
EPO	583.5 ± 109.0	1176 ± 151.6 *
G-CSF	23.3 ± 6.1	36.4 ± 9.5
GM-CSF	147.2 ± 55.7	270.6 ± 82.9
GRO/KC	261.5 ± 87.0	241.7 ± 41.5
IFN-γ	194.5 ± 46.8	681.5 ± 154.2 *
IL-1α	154.2 ± 44.2	415.4 ± 62.7 *
IL-1β	4098 ± 1179	13,083 ± 3027 *
IL-2	338.7 ± 66.4	850.5 ± 95.8 *
IL-4	196.1 ± 63.2	440.2 ± 77.0 *
IL-5	357 ± 69.4	604.1 ± 95.0
IL-6	163.2 ± 73.0	537.5 ± 98.7 *
IL-10	1161 ± 324.4	1579 ± 295.5
IL-12p70	235.4 ± 79.4	646.2 ± 128.4 *
IL-13	102.4 ± 29.5	221.7 ± 51.1
IL-17α	104.8 ± 27.4	262.9 ± 43.4 *
IL-18	3430 ± 703.7	6796 ± 920.4 *
MCSF	477.8 ± 23.0	478.3 ± 48.3
MCP-1	957.4 ± 151.4	1514 ± 212.2
MIP-3α	105.7 ± 27.2	183.3 ± 26.1
RANTES	296.2 ± 70.1	645.3 ± 114.9 *
TNF-α	155.9 ± 51.7	274.9 ± 69.8
VEGF	51.1 ± 14.8	117.4 ± 29.2

**Table 3 ijms-23-11447-t003:** The effects of AM251 and acute adiponectin treatment on skeletal muscle gene expression in HFD rats. The soleus and EDL were obtained from HFD rats treated with vehicle or AM251 for 6 weeks and exposed ex vivo to 0.1 µg/mL of adiponectin (n = 7–9) or adiponectin vehicle (n = 6–9) for 30 min. All data are presented as a mean ± SEM in arbitrary units (normalized to housekeeping genes, GAPDH, cyclophilin and β-actin.).

Gene	Skeletal Muscle	Vehicle + Adiponectin Vehicle	Vehicle + Acute Adiponectin Treatment	AM251 + Adiponectin Vehicle	AM251 + Acute Adiponectin Treatment
ADIPOR1	(a) Soleus	0.156 ± 0.119	0.095 ± 0.074	0.627 ± 0.349	0.144 ± 0.107
	(b) EDL	0.055 ± 0.014	0.037 ± 0.011	0.069 ± 0.025	0.145 ± 0.098
ADIPOR2	(a) Soleus	0.062 ± 0.018	0.061 ± 0.013	0.091 ± 0.029	0.068 ± 0.020
	(b) EDL	0.085 ± 0.037	0.086 ± 0.017	0.070 ± 0.023	0.030 ± 0.010
APPL1	(a) Soleus	0.033 ± 0.011	0.095 ± 0.048	0.413 ± 0.184	0.094 ± 0.040
	(b) EDL	0.054 ± 0.024	0.044 ± 0.009	0.112 ± 0.039	0.053 ± 0.020
APPL2	(a) Soleus	0.005 ± 0.001	0.006 ± 0.002	0.006 ± 0.003	0.006 ± 0.002
	(b) EDL	0.013 ± 0.007	0.026 ± 0.016	0.045 ± 0.031	0.011 ± 0.007
FATCD/36	(a) Soleus	0.698 ± 0.344	1.033 ± 0.330	0.373 ± 0.104	0.649 ± 0.173
	(b) EDL	0.260 ± 0.083	0.369 ± 0.104	0.627 ± 0.198	0.192 ± 0.048
AMPK	(a) Soleus	0.037 ± 0.012	0.134 ± 0.047	0.053 ± 0.026	0.098 ± 0.041
	(b) EDL	0.569 ± 0.237	0.871 ± 0.325	0.150 ± 0.055	0.165 ± 0.079
PDK4	(a) Soleus	0.663 ± 0.261	0.372 ± 0.121	0.432 ± 0.151	0.855 ± 0.415
	(b) EDL	0.141 ± 0.076	0.592 ± 0.461	0.266 ± 0.154	0.093 ± 0.037
PGC-1α	(a) Soleus	0.573 ± 0.284	0.248 ± 0.132	0.398 ± 0.157	0.685 ± 0.392
	(b) EDL	0.111 ± 0.067	0.094 ± 0.053	0.345 ± 0.120	0.077 ± 0.033

**Table 4 ijms-23-11447-t004:** Oligonucleotide PCR primers—Skeletal Muscle.

Genes	Accession Number	Forward Primer	Reverse Primer
AdipoR1	NM_207587.1	TGAGGTACCAGCCAGATGTC	CGTGTCCGCTTCTCTGTTAC
AdipoR2	NM_001037979.1	TCCATGGAGTCTCAACCTG	GGAGAGTATCACAGCCATC
AMPK subunit alpha 2 (Prkaa2)	NM_023991.1	ACTCTGCTGATGCACATGT	AGGGGTCTTCAGGAGAGG
APPL1	XR_007603	TCACTCCTTCCCCATCTTTC	TAGAGGAGGCAGCCAAAT
APPL2	NM_001108741	TGCTCGGGCTATTCACAA	AAACAGGCCCGTGACACT
β-Actin	NM_031144	CTAAGGCCAACCGTGAAA TGA	CCAGAGGCATACAGGGAC AAC
Cyclophilin	NM_017101.1	CTGATGGCGAGCCCTTG	TCTGCTGTCTTTGGAACTTTGTC
FAT/CD36	NM_031561.2	GACCATCGGCGATGAGAAA	CCAGGCCCAGGAGCTTTATT
GAPDH	NM_017008.3	AGTTCAACGCACATCAAG	GTGGTGAAGACGCCTAGA
PDK4	NM_053551.1	GGGATCTCGCCTGGCACTTT	CACACATTCACGAAGCAGCA
PGC-1α	NM_013261.3	ACCCACAGATCAGAACAAACC	GACAAATGCTCTGCTTTATTGC

AdipoR1: Adiponectin receptor 1; AdipoR2: Adiponectin receptor 2; AMPK: 5′adenosine mono-phosphate-activated protein kinase α-2; APPL1: adaptor protein, phosphotyrosine interacting with PH domain and leucine zipper 1; APPL2: adaptor protein, phosphotyrosine interacting with PH domain and leucine zipper 2; β-Actin: Beta-actin, FAT/CD36: fatty acid translocase/CD36; GAPDH: glyceraldehyde-3-phosphate dehydrogenase; PDK4: pyruvate dehydrogenase kinase 4; PGC1α: peroxisome proliferator-activated receptor gamma co-activator 1 alpha.

**Table 5 ijms-23-11447-t005:** Rat TaqMan Gene Expression Assay—Adipose.

Genes	Exon Boundary	Taqman Catalogue Number	Amplicon Length
ADRB3 (β3-AR)	2–3	Rn01478698_g1	131
CNR1 (CB_1_)	1–2	Rn00562880_m1	81
CNR2 (CB_2_)	1–2	Rn01637601_m1	68
CPT1B	11–12	Rn00682395_m1	83
HOXC9	1–2	Rn01532842_m1	94
HPRT1	8–9	Rn01527840_m1	64
IL-1β	5–6	Rn00580432_m1	74
PRDM16	5–6	Rn01516224_m1	65
SLC2A1 (GLUT 1)	8–9	Rn01417099_m1	73
SLC2A4 (GLUT 4)	9–10	Rn00562597_m1	75
TCF21	1–2	Rn01537344_m1	95
TNF-α	2–3	Rn99999017_m1	108
UCP-1	2–3	Rn00562126_m1	69

ADRB3 (β3-AR): adrenoceptor beta 3, CNR1 (CB1): Cannabinoid Receptor 1, CNR2 (CB2): Cannabinoid receptor 2, CPT1B: carnitine palmitoyltransferase 1B, HOXC9: homeobox C9, HPRT1: hypoxanthine phosphoribosyltransferase 1, IL-1β: Interleukin 1 beta, PRDM16: PR/SET domain 16, SLC2A1 (GLUT 1): Glucose Transporter 1, SLC2A4 (GLUT 4): Glucose Transporter 4, TCF21: transcription factor 21, TNF-α: Tumor necrosis factor alpha, UCP1: Uncoupling Protein 1.

## Data Availability

The data presented in this study are available on request from the corresponding author.

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
