# Peer review of "CB1 Ligand AM251 Induces Weight Loss and Fat Reduction in Addition to Increased Systemic Inflammation in Diet-Induced Obesity"

_ijms, 2022, doi:10.3390/ijms231911447_

Round 1
Reviewer 1 Report
O’Keefe et al. investigated the effect of the compound AM251, an agonist/antagonist of cannabinoid receptor 1, on diet-induced obese (DIO) rats. They observe that AM251 induces weight loss and fat reduction in the DIO rat model, however, importantly, they also observe that AM251 significantly alters several plasma hormones and an array of pro-inflammatory cytokines representative of whole-body inflammation.
The inherent question, the experimental model, methods used, results, analysis and interpretation of data and conclusions are reasonable and acceptable to me.
However, I have a few comments – addressing them could improve the strength of the manuscript:
1. One advantage of using animal models is that, in addition to allowing molecular level studies, it also allows tissue/cell level studies/observations. I wonder why the authors did not consider providing pictures of tissues that show significant weight reduction. Also, some pictures at the histological level of all types of tissues examined could be included in the manuscript. Sometimes, as we know, ‘a picture is worth a thousand words’.
2. The authors examined expression level of several genes by PCR, many of which show significant change in expression level. No protein-level experiments supporting PCR-level results have been carried out by the authors. I would consider IHC-type of experiments for at least a few of those genes in the relevant tissues between the groups of animals.
3. A picture of the various blood cells at various times (and at various resolutions) with and without AM251 could also be informative. Consider evaluating relative abundance of various blood cells at different times of treatment.
4. I would also advise creating a plot of body weight vs time between the two groups of rats.
5. Also consider creating a plot for food consumption vs time between the two groups.
Author Response
Please see the attached response to the reviewers

Reviewer 2 Report
O´Keefe and Coworkers demonstrate weight loss and expected and unexpected alterations caused by CB1 ligand AM251. Most outstandingly unexpected finding is the marked increase in systemic inflammation markers, which makes the data highly important to be published.
The authors discuss that tissue remodeling may underly the inflammatory response. Please give the body weights at start of the treatment period! Is there a weight loss or just reduced weight gain, which may not be regarded as tissue remodeling?
To get hints on the source of inflammation, inflammatory gene expression in the fat depots should be analyzed.
To be discussed:
Is there any data on CB1 knockout (or CB1-ligand treated) animals on systemic inflammatory markers in obesity models?
What are the differences between Rimonabant and AM251? Is there a possible explanation for the different results by these differences?
Minor:
I suggest to present the data of the tables as diagrams to facilitate reading.
Author Response
Please see attached response to the reviewers

Round 2
Reviewer 2 Report
1. OK
2. I don´t see the point why the authors do not want to include expression of inflammatory genes to their analyses. Addition of information, where the unexpected inflammatory response originates (or not) would be crucial.
3. OK
4. Please consider to add this issue to the discussion of the paper.
5. OK
Author Response
2. I don´t see the point why the authors do not want to include expression of inflammatory genes to their analyses. Addition of information, where the unexpected inflammatory response originates (or not) would be crucial. |
Thank-you for the continual feedback. We have now completed analysis of IL-1β, IL-6 and TNF-α in the various adipose tissue. This additional analysis has been included in Figures 3, 4, 5. The discussion of this new data is now included in lines 296-308 of the manuscript. |
4. Please consider to add this issue to the discussion of the paper. |
This has now been included in the manuscript lines 309-324. |
Round 3
Reviewer 2 Report
Thanks to the authors for addressing all comments adequately.